Impact of accelerated aging on seed quality, seed coat physical structure and antioxidant enzyme activity of Maize (Zea mays L.)

Satya Srii Vijayan
Nagarajappa Nethra nethraharsha@gmail.com
Department of Seed Science and Technology, University of Agricultural Sciences , Bangalore , Karnataka , India
Uversky Vladimir
Electronic publication date: 2024 Nov 11
Publication date: 2024
Volume: 12
Electronic Location ID: e17988
Received 2024 May 17; Accepted 2024 Aug 7
Copyright: ©2024 Satya Srii and Nagarajappa
Copyright year: 2024
Copyright holder: Satya Srii and Nagarajappa
License: This is an open access article distributed under the terms of the Creative Commons Attribution License, which permits unrestricted use, distribution, reproduction and adaptation in any medium and for any purpose provided that it is properly attributed. For attribution, the original author(s), title, publication source (PeerJ) and either DOI or URL of the article must be cited.
License URL: https://creativecommons.org/licenses/by/4.0/

Keywords: Catalase, Deterioration, Epidermis, Free radical, Microsculpture, Peroxidase, Polyphenol oxidase, Superoxide dismutase, Viability

Funding: TRuCapSols, Bethlehem, USA to Nethra Nagarajappa project Ab/Ac 5371 Understanding the biochemical composition of seed coat in different crops to develop biodegradable membrane for encapsulation Department of Science and Technology, India for providing INSPIRE fellowship to V. Satya Srii The expense of this research was funded by TRuCapSols, Bethlehem, USA to Nethra Nagarajappa under the project Ab/Ac 5371, Understanding the biochemical composition of seed coat in different crops to develop biodegradable membrane for encapsulation. The Department of Science and Technology, India provided an INSPIRE fellowship to Satya Srii Vijayan. The funders had no role in study design, data collection and analysis, decision to publish, or preparation of the manuscript.

==============================
Aging induces many deteriorative changes to seeds during storage like protein degradation, enzyme inactivation and loss of membrane integrity. In this study, we investigate the impact of accelerated aging on seed quality, seed coat physical structure and antioxidant enzyme activity of maize. Three genotypes African Tall, MAH 14-5 and a local landrace were selected and artificially aged for 96 and 120 h. The aging process led to a decrease in germination, vigour, and total dehydrogenase in seeds, while the electrical conductivity of seed leachates increased, indicating a decline in seed quality. Additionally, there was a variation in the microsculpture pattern of seed coats between genotypes. There was an accumulation of damage on the seed coat surface as the seeds aged and higher damage occurred in African Tall followed by MAH 14-5 and local landrace. Higher catalase, superoxide dismutase, peroxidase and polyphenol oxidase activity were reported in the seed coat of Local landrace and MAH 14-5 that resisted aging and deterioration while, lower catalase, superoxide dismutase, peroxidase and polyphenol oxidase activity was reported in African Tall seed coat that deteriorated during aging. Decrease antioxidant activity in aged seeds might be a possible cause of seed deterioration due to the accumulation of free radicals. Thus, these results clearly show the influence of seed coat structure and antioxidant activity on seed quality during aging.

Introduction

Seed aging is an irreversible and inexorable process of a progressive decrease in vigour ultimately leading to the loss of seed viability (Stewart & Bewley, 1980; Lehner et al., 2008; Arc et al., 2011). The rate of seed aging depends upon the genotype/species, the conditions prevailing during storage like moisture content, temperature, humidity and seed composition (Roberts, 1973). Accelerated aging is used extensively to study the mechanism of aging and the associated deterioration process in seeds. In accelerated aging, seeds are exposed to high temperatures and high relative humidity which hastens the deteriorative process in seeds (Delouche & Baskin, 1973). At first, this was proposed as a method to evaluate seed storability; later, it was used as a rapid technique to study relative seed storability. From then accelerated aging was used to investigate the changes that occur in seeds during deterioration.

Studies have extensively reported the physical, biochemical, physiological and genetic changes in seeds resulting from accelerated aging. Studies show that the loss of seed membrane integrity results in increased seed leachate, which is measured by the electrical conductivity of artificially aged seeds (Basra et al., 2003). Also, studies show an increase in free fatty acid and lipid peroxidation of aged seeds (Bailly et al., 1998; Basra et al., 2003). Furthermore, the antioxidant defense mechanism in seeds has also been impacted, with studies indicating decreased activities of catalase, peroxidase, and glutathione reductase (Bailly et al., 1998; Lakshmi, Jijeesh & Seethalakshmi, 2021). DNA alterations and programmed cell death (PCD) were reported in aged seeds (El-Maarouf-Bouteau et al., 2011). Also, protein carbonylation resulting from reactive oxygen species (ROS) production was reported in aged seeds (Lilya, Ouzna & Réda, 2021). Though there are numerous studies about changes brought about by aging in seeds (Pukacka & Ratajczak, 2007; Demirkaya, Dietz & Sivritepe, 2010; Brar, Kaushik & Dudi, 2019; Nagel et al., 2019), studies are scarce concerning changes in the seed coat. Studies about seed coat mainly report the seed coat color and thickness related to seed quality during deterioration by artificial aging (Kuchlan, Dadlani & Samuel, 2010). However, the knowledge about changes in the physical structure and antioxidant enzyme activity of seed coat during artificial aging is very little.

The seed coat, or testa, is the outer covering of a mature seed containing an embryo and nutritive tissue. It controls the exchange of gases between the embryo and its environment, protecting it from mechanical damage, pests, and diseases (Mohamed-Yasseen et al., 1994; Weber, Borisjuk & Wobus, 1996). Seed coats are the primary defense of seeds acting as modulators of seed-environment relationships. The physical structure of the seed coat indirectly prolongs storability by influencing permeability. Other than lignification, the properties of seed coats like texture and color were found to influence permeability (Calero, West & Hinson, 1981; Yaklich & Barla-Szabo, 1993; Ragus, 1987; Qutob et al., 2008). It was reported that porous seed coats were more permeable than the non-porous seed coats in soybean (Calero, West & Hinson, 1981; Yaklich & Barla-Szabo, 1993). Also, a recent study by Satya Srii, Nagarajappa & Vasudevan (2022) in soybean showed that physical structural changes in seed coat induced by accelerated aging led to seed quality loss and imbibition injury.

Concerning reactive oxygen species (ROS), oxygen is a highly toxic molecule. Although it is slightly reactive on its own, it has the potential to give rise to highly reactive and potentially harmful free radicals during the electron transport chain (Bailly, 2004). Therefore, the role of antioxidant enzymes in maintaining the optimal oxidative state inside the seed is critical in determining its storability. Studies in Caesalpinia pulcherrima and Delonix regia showed that seed coats contained more antioxidants compared to embryos and endosperms (Sittikijyothin et al., 2014). Brazilian and Peruvian bean cultivars contained more antioxidants and phenolics in seed coats than in cotyledons (Ranilla, Maria & Lajolo, 2007). Thus studying the changes in antioxidant enzyme activity especially in seed coat during aging will offer insights into the influence of seed coat antioxidants in aging and seed deterioration.

We hypothesized that accelerated aging would impact the physical structure of the seed coat and its antioxidant activity. To test this hypothesis, our study aims to investigate the effects of accelerated aging on the seed quality, seed coat physical structure, and antioxidant enzyme activity of maize (Zea mays L.). The proposed study focuses on three genotypes of maize: African Tall (white seed coat), MAH 14-5 (orange seed coat), and a local cultivar (red seed coat), selected based on their seed coat color. This crop is of interest due to its good storage capabilities, which creates an opportunity to investigate the mechanisms that enhance seed storability. Additionally, the presence of different color genotypes allows for the exploration of differences in seed coat structure and composition concerning seed coat color. Previous studies have suggested that cultivars with darker seed coats in chickpeas (Gvozdeva & Zhukova, 1971), soybeans (Shahi & Pandey, 1982), snap beans (Phaseolus vulgaris) (Prasad & Weigle, 1976), and French beans (Powell, Oliveria & Matthews, 1986) are less permeable and have a longer lifespan compared to cultivars with lighter seed coat colors. Below are the objectives of the study.

1. To know the effect of aging on the seed quality.

2. To know the physical structure and histochemical analysis of seed coat in artificially aged seeds by microscopic techniques.

3. To evaluate the antioxidant enzyme activities in seed coat of artificially aged seeds by analytical techniques.

Material and Methods

Seed material

Fresh seeds of African fodder maize and MAH 14-5 were received from the Seed Stores, National Seed Project, University of Agricultural Sciences, Bangalore and local landrace genotype was received from farmers in Tamil Nadu. The obtained seeds were multiplied during Kharif season in the fields of AICRP on Seed Technology, National Seed Project, Gandhi Krishi Vignan Kendra, University of Agricultural Sciences, Bangalore for the experiments.

Accelerated aging

Artificial aging was performed according to ISTA guidelines (ISTA, 2013). The moisture content of the fresh seed samples was determined by the hot air oven method (ISTA, 2013) before proceeding with accelerated aging. To simulate artificial aging (AA), plastic boxes were first sterilized with 5% sodium hypochlorite, dried, and then filled with 40 ± 1.0 ml of distilled water. Each box contained forty-two grams of seeds laid out in a single layer to ensure consistent moisture uptake from the humid environment. The lid was then placed on each box. These boxes were then placed on the shelves of the aging chamber (Thermo Fisher Scientific, Model: IGS 60/100/180; Thermo Fisher Scientific, Waltham, MA, USA) with 2.5 cm of space between them to ensure even temperature distribution. The temperature was set at 41 ± 0.3 °C with 100% relative humidity and was regularly monitored. The study investigated seed germination during accelerated aging at 24, 48, 96, and 120 h. The results indicated a decrease in germination percentage starting at 96 h. As a result, the effects of seed deterioration were studied using the 96 and 120-hour accelerated aging treatments. After 96 and 120 h of aging the AA boxes were removed from the chamber. The seed moisture content of seeds after aging was measured using a digital moisture meter.

Seed quality

Germination percent and viability

Seed germination test was performed within one hour after removal from the aging chamber for 100 seeds in four replicates by between paper method at 25 °C for each aging treatment and control (fresh, non-aged seeds). The first and final count was taken on the 4th and 7th day for maize and the seedlings were evaluated as per International Seed Testing Association guidelines (ISTA, 2013) considering the percentage of normal seedlings to be germinated.

The seed viability testing was conducted following the procedures outlined in the ISTA (International Seed Testing Association) guidelines from 2013 (ISTA, 2013). This involved testing 50 seeds in 2 replicates for each aging treatment using Tetrazolium (Tz) staining. The process included soaking the seeds in distilled water for 18 h, followed by cutting them longitudinally through the embryo and 3/4th of the endosperm. Subsequently, the seeds were soaked in a 1% Tz solution for 2 h at 30 °C in the dark. Finally, the uptake of stains was evaluated according to the International Seed Testing Association guidelines.

Seedling vigour indices

To calculate the seedling vigour index-I (SVI-1), the seedling length (root and shoot) of five randomly selected seedlings in each replicate was measured while taking the final count of seeds in the germination test. The vigour index was calculated by multiplying germination (%) and seedling length (cm). The lot showing higher values was highly vigorous (ISTA, 2013). Seedling vigour index I=Germination percent×Seedling length.

To calculate the seedling vigour index-II (SVI-II), the seedling dry weight of five randomly selected seedlings in each replicate was measured while taking the final count of seeds in the germination test. The vigour index was calculated by multiplying germination (%) and seedling dry weight. For dry weight determination, the seedlings are removed and dried in hot air oven at 100 °C for 24 h. The lot showing higher values was highly vigorous. Seedling vigour index II=Germination percent×Seedling dry weight.

Electrical conductivity

Membrane deterioration was measured by an electrical conductivity (EC) test as per the modified ISTA method (ISTA, 2013) and as described by Satya Srii, Nagarajappa & Vasudevan (2022). Fifty seeds in two replicates in each aging treatment along with control were soaked in 250 ml of distilled water for 24 h at 20 °C and the conductivity of the elute was measured using a conductivity meter (Model: 306; Systronics, Vadaj, India).

Seed moisture content

The seed moisture content of seeds was measured before and after aging using a seed moisture meter (Model: aGrain; Make: AG-07). The seed moisture meter method was used for moisture estimation as it is a non-destructive method and as the seeds need to be used for other analyses after moisture estimation.

Total dehydrogenase

The total dehydrogenase activity was measured by the method suggested by Kittock & Law (1968). The seeds imbibed in Tz viability testing were washed thoroughly with distilled water and the red-colored formazan from the stained regions was extracted by soaking with 5 ml of 2-methoxy ethanol for 8 h in an airtight container. The extract was decanted and the color intensity was measured in a spectrophotometer (model: 2301; SICAN) at 480 nm with a suitable blank (Methoxy ethanol). The total dehydrogenase activity (TDH) was expressed as absorbance. The significance of values for different sample groups of three genotypes were analyzed using ANOVA (with single factor) with post-hoc Tukey Krammer test using SPSS software.

Microscopy

Light microscopy

Seed coat sectioning was obtained by inhouse technique i.e., nut and bolt method as described by Satya Srii, Nagarajappa & Vasudevan (2022). The seed was embedded in wax and placed inside a nut and sections were taken using a blade. Care was taken not to create damage to seed coats during sectioning by embedding the seed coats in wax for sectioning. Seed coat sections of five to ten seeds in each aging treatment along with control were examined under the light microscope (0807875; CETA).

Scanning electron microscopy

The seed coat sections from the dorsal surface were first dehydrated with various concentrations of ethanol and then the samples were coated with gold particles. The surface features of dry seed coats were then examined with a Zeiss scanning electron microscope (EVO-18; Zeiss) at 15 kV as described by Ma et al. (2004). For each genotype, four seeds were included for analysis.

Histochemical studies

Polycarboxylic acid and phenolic acid

The seeds were soaked in water for one day before being sectioned. After that, the seeds were embedded in wax blocks and sections were obtained using a microtome at Prakash Laboratories in Bangalore. The sections were then fixed to slides for further staining. The detection of polycarboxylic acids and phenolic compounds was carried out using the method described by O’Brien, Feder & Mccully (1964) and Vu, Veluswamy & Park (2014). Dehydrated tissues were stained with a 0.05% toluidine blue solution. The solution was applied to the slide with the specimens, and the samples were stained for five minutes. The excess stain was removed by dipping the slides into two changes of distilled water for three minutes. The tissues were then covered with a cover glass and observed under a light microscope. As a result, the polycarboxylic acid and phenolic acid in the seed coat appeared blue after staining.

Antioxidant enzyme activity

Seed coat tissue (0.5 g) was ground to a fine powder with a mortar and pestle using liquid nitrogen. Potassium phosphate buffer (2 ml), (50 mM; pH 7.0) containing EDTA (1 mM), and soluble PVP (1%) was added and homogenized. The homogenate was transferred to 2ml Eppendorf tubes and centrifuged (Eppendorf, India) at 12,000 rpm for 15 min at 4 °C and the supernatant was carefully transferred to fresh tubes and used for estimation of antioxidant enzymes, viz., catalase (CAT), peroxidase (POX), and polyphenol oxidase using Multiskan Thermo scientific spectrophotometer. Three independent replications of enzyme extract were maintained for enzyme assays.

Catalase activity

The enzyme activity was assayed according to Masia (1998) with minor modifications. The assay mixture of catalase activity contained 165 µl of 0.05 M Phosphate buffer pH (7.0), with 20 µl of 0.3% H2O2 and 15 µl of enzyme extract. The change in absorbance was recorded at 240 nm for 6 min at an interval of every minute with a Multiskan spectrophotometer (Thermo Fisher Scientific). The results were expressed as a change in µmoles of H2O2 decomposed mg−1 of protein min−1. Catalase activityUnits/gm FW=OD change/min×FactorThe volume of a sample×Total volumeWt. of sample.

Factor: 1 OD change/min = 187.528 Units of the enzyme.

Peroxidase activity

An assay of peroxidase activity was carried out by the procedure described by Subhas Chander (1990). The assay mixture of peroxidase contained 124 µl of 0.05 M Phosphate buffer pH (7.0), with 13 µl of catechol and 13 µl of H2O2 and 50 µl enzyme extract. The change in absorbance was recorded at 450 nm for 6 min at an interval of every minute with a Multiskan spectrophotometer (Thermo Fisher Scientific). The results were expressed as a change in Units/gm fresh weight. Peroxidase activityUnits/gm FW=OD change/min∗Factorml of sample taken for assay∗Total volumewt. of sample.

OD change/min = 0.033 Units of enzyme.

Superoxide dismutase activity

An assay of superoxide dismutase activity was carried out by the procedure described by Du & Bramlage (1994). The assay mixture of superoxide contained 5 µl of 0.05 M Phosphate buffer pH (7.0), 2 µl of EDTA, 67 µl of methionine, 60 µl of enzyme extract, 33 µl of riboflavin and 33 µl of Nitroblue tetrazolium. Two groups of assay mixture from each sample of prepared and one was incubated in the light and the other in dark condition for 30 min. The change in absorbance was recorded at 560 nm with a Multiskan spectrophotometer (Thermo Fisher Scientific). The results were expressed as a change in Units/mg of protein. Superoxide dismutase activityUnits/gm FW=Aml of sample taken for assay∗Total volumewt. of sample.

Polyphenol oxidase activity

An assay of polyphenol oxidase activity was carried out by the procedure described by Kumar & Khan (1982). The assay mixture of polyphenol oxidase contained 144 µl of 0.05 M Phosphate buffer pH (7.0), with 6 µl of 1.25% pyrogallol and 50 µl of enzyme extract. The change in absorbance was recorded at 450 nm for 6 min at an interval of every minute with a Multiskan spectrophotometer (Thermo Fisher Scientific). The results were expressed as a change inAbs/min/gm fresh weight. Polyphenol oxidaseAbs/min/gm FW=OD change/minuteml of sample taken for assay∗Total volumewt. of sample.

Estimation of polyphenols

The estimation of polyphenols was done using the protocol described by Xu & Kow-Ching (2007) and Nagaralli (2015). The defatted sample of 0.5-1 g of soybean and maize seed coat powder was weighed and ground with a pestle and mortar in ten-time volume of 80% ethanol. The homogenate was centrifuged at 10,000 rpm for 20 min. The supernatant was saved. The residue was re-extracted with five times the volume of 80% ethanol, centrifuged and supernatants pooled. The supernatant was evaporated to dryness. The residue was dissolved in a known volume of distilled water (5 ml). The different aliquots (0.2-2 ml) were pipetted out in test tubes. The volume was made up to 3 ml with water in each tube 0.5 ml of Folin-Ciocalteau reagent was added (standard). After 3 min, 2 ml of 20% Na2CO3solution was added to each tube and mixed thoroughly. The tubes were placed in boiling water for exactly 1 min, cooled and absorbance measured at 650 nm against a reagent blank. A standard curve was prepared using different concentrations of catechol.

Data analysis

Descriptive statistics were performed using Microsoft Excel 2010. The significance of the difference in seed quality values between genotypes before and after storage was analyzed individually using Microsoft Excel 2010. Post-adhoc tests were performed using SPSS software.

Results

Seed quality after aging

Seed viability and germination

To investigate the impact of accelerated aging, we chose aging treatments of 96 and 120 h after establishing viability at different aging intervals. This allowed us to work with seeds that were still viable but had reduced vigour, representing the initial stages of deterioration. The goal was to examine structural changes in the seed coat of viable seeds due to aging. Our results indicated no difference in seed viability after aging across three different genotypes, with all seeds maintaining 100% viability. However, the germination rate varied among genotypes and within the aging times of 96 and 120 h (see Table 1).

The initial germination per cent was 100 per cent for all genotypes while after 120 h of aging it was 90, 92 and 96 per cent in African tall, MAH 14-5 and Local landrace respectively. A significant difference in seed germination per cent after aging was observed and also germination per cent between genotypes differed significantly with p < 0.01. However, the Local landrace and MAH 14-5 showed less deterioration reflected as higher germination per cent after aging than African Tall. This comparatively reduced loss of deterioration in maize seeds with colored seed coat may be attributed to the presence of phenols in the seed coat which offer seed coat strength once oxidized (Pourcel et al., 2005) and is also rich in antioxidants (Madrera et al., 2021).

Seedling vigour indices

Results showed that before aging vigour indices (SV-I and SV-II) did not significantly differ among the three genotypes however after aging local landrace seeds had higher vigour indices with 2764 and 65.66 as SV-I and SV-II, respectively followed by MAH 14-5 with 2222 and 50.32 as SV-I and SV-II, respectively and least in African tall with 1929 and 42.39 as SV-I and SV-II, respectively (Table 2). A similar trend was observed in seed germination per cent while, there was no change in viability.

Electrical conductivity of seed leachates

To confirm the differential response of genotypes related to the deterioration of seeds due to aging, the damage to seed coat after accelerated aging between three genotypes was measured as electrical conductivity (EC) of seed leachates. The EC in three genotypes increased with aging time but there was a difference in EC values between genotypes (Table 3). Local red landrace had comparatively low EC than MAH 14-5 and African tall both before and after aging. Local landrace, MAH 14-5 and African tall had ECs of 5.94, 6.93 and 11.04 µScm−1g−1, respectively, in fresh seeds, while 8.82, 9.65, 15.09 in seeds aged for 120 h.

Seed moisture content

The initial moisture content of seeds before aging in all three genotypes was 10 per cent. However, the moisture content of seeds after aging showed that African tall had higher moisture content (16 and 17 per cent after 96 and 120 h of aging) followed by MAH 14-5 (14 and 15 per cent after 96 and 120 h of aging) while Local landrace had least moisture increase of 11 and 12 per cent after 96 and 120 h of aging, respectively (Table 4). This differential moisture absorption during aging might be a possible reason for the difference in seed quality among genotypes after aging.

Table 1 Seed germination and viability measured before and after aging in three maize genotypes.

Genotypes	Seed germination (%)	Seed Viability (%)	
	Control	Aging (96 h)	Aging (120 h)	Mean	Control	Aging (96 h)	Aging (120 h)	
African tall	100 ± 0.00a	94 ± 1.15b	90 ± 0.75c	96.91	100 ± 0.00a	100 ± 0.00a	100 ± 0.00a	
MAH 14-5	100 ± 0.00a	96 ± 0.5b	92 ± 0.5c	96.16	100 ± 0.00a	100 ± 0.00a	100 ± 0.00a	
Local landrace	100 ± 0.00a	99 ± 0.81b	96 ± 0.5c	98.58	100 ± 0.00a	100 ± 0.00a	100 ± 0.00a	
Notes.

Mean ± SD for replicated experiments are shown.

Different letters in superscript indicate significance at p < 0.01 between different aging periods.

Table 2 Seed Vigour Index-I measured before and after aging in three maize genotypes.

Genotypes	Seed Vigour Index-I	Seed Vigour Index-II	
	Control	Aging (96 h)	Aging (120 h)	Control	Aging (96 h)	Aging (120 h)	
African tall	3038 ± 1.2a	2479 ± 1.75b	1929 ± 0.78c	71.50 ± 0.58a	56.11 ± 0.79b	42.39 ± 1.14c	
MAH 14-5	3148 ± 0.95a	2631 ± 1.15b	2222 ± 1.7c	72.10 ± 0.89a	66.81 ± 0.41b	50.32 ± 0.14c	
Local landrace	3249 ± 0.54a	2945 ± 0.78b	2764 ± 1.25c	72.40 ± 0.74a	70.48 ± 0.48b	65.66 ± 0.59c	
Notes.

Mean ± SD for replicated experiments are shown.

Different letters in superscript indicate significance at p < 0.01 between different aging periods.

Table 3 Electrical conductivity of seed leachates measured before and after aging in three maize genotypes.

Genotypes	Electrical conductivity (µS cm−1 g−1)	
	Control	Aging (96 h)	Aging (120 h)	
African tall	11.04 ± 0.76a	12.70 ± 0.42b	15.09 ± 0.55c	
MAH 14-5	6.93 ± 0.72a	8.97 ± 0.34b	9.65 ± 0.44c	
Local landrace	5.94 ± 0.51a	7.67 ± 0.41b	8.82 ± 0.22c	
Notes.

Mean ± SD for replicated experiments are shown.

Different letters in superscript indicate significance at p < 0.01 between different aging periods.

Table 4 Seed moisture content measured before and after aging in three maize genotypes.

Genotypes	Seed moisture content (%)	
	Control	Aging (96 h)	Aging (120 h)	
African Tall	10 ± 0.00a	16 ± 0.15b	17 ± 0.08c	
MAH 14-5	10 ± 0.00a	14 ± 0.08b	15 ± 0.55c	
Local landrace	10 ± 0.00a	11 ± 0.25b	12 ± 0.15c	
Notes.

Mean ± SD for replicated experiments are shown.

Different letters in superscript indicate significance at p < 0.01 between different aging periods.

Total dehydrogenase activity (TDH)

Results showed that fresh local landrace seeds (0.82) had comparatively high TDH than MAH 14-5 (0.65) and African tall (0.27). During aging TDH content in all three genotypes significantly decreased with aging time (Fig. 1). However, the amount of TDH in seeds aged for 120 h is comparatively still higher in local landrace than MAH 14-5 and African tall.

Figure 1 Changes in total dehydrogenase levels (OD @ A480nm) in three maize genotypes. C, fresh seeds; T1, 96 h of ageing; T2, 120 h of ageing.

Black bars indicate the standard error. Different letters above the bar denotes that TDH levels differed significantly between control and aged seeds at P < 0.05.

Changes in the physical structure of the seed coat

Light microscopy

The sections showed three layers of seed coat ie. epidermis, hypodermis and interior parenchyma (Fig. 2).

Figure 2 Seed coat section of three genotypes of maize (A) African Tall, (B) MAH 14-5, (C) local red landrace, showing three layers of the seed coat.

E, Epidermis; H, Hypodermis; IP, Inner most layer; En, Endosperm. Top panel shows free-hand unstained sections observed under light microscope and the bottom panel shows sections microtome sections stained with hematoxylin and eosin stain which stains the cells. Black arrows shows the difference in cell type and arrangement in hypodermis. Scale bar: 20 µm.

In addition to the above, the sections from our study revealed that there was no separate cuticle layer to be present in the maize seed coat. The top panels (Fig. 2) shows that between three genotypes, the interior parenchyma layer is prominently present in MAH 14-5 and local landrace while it is crushed in African Tall. The innermost layer of MAH 14-5 and Local landrace had well-differentiated cells while it was not well-differentiated/crushed in the African Tall genotype.

Besides the difference in the interior parenchyma layer, the bottom panels of Fig. 2 showing sections stained with hematoxylin and Eosin stain reveals the difference in cell types in the hypodermis. African tall had irregularly circular cells while MAH 14-5 had elongated elliptical cells and local landrace had very closely arranged lines of rectangular cells in hypodermis. Also, Fig. 3A & 3B show intact hypodermis in fresh African Tall seeds while, it shows the distorted cells in hypodermis in 120 h aged African Tall Seeds. Results of EC show 120 h aged seeds to have high EC while fresh seeds have less permeability and less EC in African Tall. Thus this also proves the possible role of hypodermal cells in seed coat damage.

Figure 3 Structure of seed coat of African Tall seeds under light microscope. (A) Cross-section of the seed coat of fresh African Tall seed. (B) Cross-section of the seed coat of 120 h aged African Tall observed under the light microscope.

The black arrow in (A) shows the intact hypodermis in fresh African Tall seed while in (B) it shows the distorted cells in hypodermis in 120 h aged African Tall Seeds. Scale bar: 20 µm.

Scanning electron microscopy revealed that the seed coat microsculpture was distinct in each genotype. Seed coat microsculptures are the primary superficial sculptures that are related to the shape and arrangement of epidermal cells of the seed coat. Figure 4 shows that African Tall seeds have a slightly curved rectangular pattern, MAH 14-5 had straight bordered rectangles while, local landrace had a curved line pattern on the epidermis. SEM images also revealed that intact seed coat in fresh seeds of three genotypes had no damage; however, the damage in seed coat started to accumulate as the seeds aged (Fig. 5). The level of seed coat damage was least in aged seeds of Local landrace followed by MAH 14-5 while African tall incurred increased damage. However, Fig. 6 shows clearly that the pattern of damage is concerning the microsculpture pattern where the damage is along the microsculpture pattern.

Figure 4 Seed coat micro sculpture pattern in epidermis among three maize genotypes. (A) African Tall. (B) MAH 14-5. (C) local red landrace.

Panel (A) has slightly curved rectangles, (B) has straight bordered rectangles and (C) has curved line pattern on epidermis. Scale bar: 100 µm.

Figure 5 Seed coat damage in epidermis among three maize genotypes: African Tall, MAH 14-5 and local red landrace.

Control, fresh seeds; T1, 96 h aged seed T2, 120 h aged seed. Yellow arrow indicates the damage in seed coat epidermal layer. Scale bar for local red landrace control: 100 µm and other images: 20 µm.

Figure 6 The pattern of damage in seed coat epidermis in (A) African Tall, (B) MAH 14-5, (C) local red landrace.

Yellow arrows indicate the damaged area in seed coat epidermis.

Phenol staining in the seed coat

The seed section was stained with hematoxylin and Eosin stain which is used to study the structure of cells and arrangements. Figure 7A shows the detailed structure and arrangement of cells in maize seed coat. As the difference in phenols was predominant between genotypes shown quantitatively by phenol estimation, their presence in a specific layer of the seed coat was studied by staining the seed coat for phenol compounds. Staining with Toluidine blue (Fig. 7B) revealed that phenols were present only in the epidermal and hypodermal layers of the seed coat while it was not present in the interior parenchyma layer.

Figure 7 Structure and histochemistry of maize seed coats.

(A) Seed section of local red landrace showing seed coat (SC) layers. E, Epidermis; H, Hypodermis; IL, Interior Layer with parenchyma cells (B) Seed coat section of Local Red Landrace stained with Toulidine blue which stains phenols. Red arrow indicate the blue staining of epidermal layer showing phenol accumulation. Scale bar: (A) and (B)—20 µm.

Antioxidant activity in seed coats

Catalase

Results of catalase estimation (Fig. 8A) in seed coat showed that fresh seed coat had comparatively higher catalase activity than aged seed coat with catalase activity getting reduced with increased aging time. However, the fresh seeds of local landrace and MAH 14-5 had higher catalase (19.95 and 18.3 µmoles of H2O2 decomposed mg−1 of protein min−1respectively) than African Tall genotype with 16.15 µmoles of H2O2 decomposed mg−1 of protein min−1. Also, the level of decrease in catalase activity in aged seeds was lower in colored seed coat than in colorless seed coat.

Figure 8 Antioxidant enzyme activity in seed coat of maize genotypes (African Tall, MAH 14-5 and local landrace) before and after ageing.

(A) Catalase, (B) Superoxide dismutase, (C) peroxidase, (D) polyphenol oxidase. Black bars indicate standard error. Different letters above bar indicate significant difference between ageing time at p < 0.01.

Superoxide dismutase (SOD)

Results of superoxide dismutase estimation (Fig. 8B) in seed coat showed that fresh seed coat had comparatively higher SOD activity than aged seed coat with SOD activity getting reduced with increased aging time. However, the fresh seeds of local landrace and MAH 14-5 had higher SOD (23.32 and 20.3 units per mg of protein, respectively) than the African Tall genotype with 19.15 units per mg of protein. Also, the level of decrease in SOD activity in aged seeds was lower in genotypes with colored seed coats than in colorless seed coats.

Peroxidase

Results of peroxidase activity (Fig. 8C) showed that seed coat of African Tall had comparatively higher peroxidase than seed coats of Local landrace and MAH 14-5 in fresh seeds and at all points of aging. Also, the peroxidase activity in seeds decreased as seeds aged with 16.15 to 10.32 µmoles/ min/g fresh weight in fresh and 120 h aged seeds respectively in African Tall, while from 18.20 to 11.67 µmoles/ min/g fresh weight in MAH 14-5 and 20.03 to 13.32 µmoles/ min/g fresh weight in fresh and 120 h aged seeds, respectively.

Polyphenol oxidase

Results of polyphenol oxidase (Fig. 8D) show that fresh African Tall seed coats had significantly lower polyphenol oxidase (0.06 units per g fresh weight) than MAH 14-5 and local landrace seed coats (0.16 and 0.18 units per g fresh weight, respectively). The polyphenol oxidase activity showed no significant changes with aging times in all genotypes. The enzyme polyphenol oxidase (PPO) is shown to catalyze the oxidation of phenolic compounds which in turn impart dark coloration and cause post-harvest browning (Araji et al., 2014). Thus, colored seed coats of local landrace and MAH 14-5 had higher polyphenol oxidase than white seed coats of African Tall.

Phenols in seed coat

Local landrace seed coat had a higher content of phenol (9.801  ±  0.01 mg/g) followed by MAH 14-5 (6.693  ±  0.27 mg/g) and least in African Tall seed coat (1.184  ±  0.3 mg/g). The content of phenols among genotypes showed significant differences at p < 0.01 in ANOVA with a single factor.

Discussion

Seed quality after aging

The results of viability and germination tests showed that though three genotypes were 100% viable had a decrease in germination from an initial 100 per cent in control (fresh) seeds. These contrasting results between viability and germination after aging between genotypes are because of the increased fraction of abnormal seedlings produced by aged seeds which would be considered not germinated in germination tests as per ISTA test guidelines. The genotypes though had 100% viability even after aging, the deteriorative changes that occurred due to aging were reflected as abnormal seedlings. The variation in seed vigor between different genotypes after aging may be attributed to differences in germination rates, even though both genotypes are 100% viable. Vigor estimation takes into account germination rates for its calculation, so any abnormal seedlings will result in varied germination rates in aged seeds, despite the seeds being otherwise viable. Results showed that seeds with a colored seed coat ie., local landrace and MAH 14-5 had comparatively higher vigour after aging than seeds with a white seed coat ie., African Tall. Studies have already reported low vigour due to increased deterioration and damage in genotypes with colorless seed coat than in colored genotypes which have been related to the seed coat composition and thickness (Kazim, 2010).

The results showed that EC values increased as seeds aged. Parrish, Leopold & Hanna (1982) reported that an increase in electrolyte leakage in deteriorated seeds is an indication of membrane deterioration leading to the aging of seeds. However, local landrace and MAH 14-5 had lower EC than African Tall revealing that three genotypes had a different level of membrane damage during aging, with Local landrace (red) having less seed coat damage followed by MAH 14-5 (orange) and African Tall (white). The reason for the higher integrity of seed coat in colored genotypes could be due to the presence of proanthocyanidins with free radical scavenging activity (Takahata, Ohnishi-Kameyama & Furuta, 2001) which would in turn help in cell repair mechanisms preventing membrane damage (Bailly, 2004). The results of TDH correlated with vigour and germination per cent of seeds. African tall seeds during aging had reduced germination, vigour and TDH while local landrace and MAH 14-5 during aging showed comparatively higher vigour, germination and TDH. Bam et al. (2006) showed in their study that high-vigour rice seeds had higher TDH activity compared to low-vigour seeds. This is in line with the results of higher TDH activity in Local landrace which had higher vigour than African Tall with low TDH and vigour. Previous studies showed that TDH content decreases in aged and deteriorated seeds (Verma, Verma & Tomer, 2003).

Besides these, the differential moisture absorption during aging may be attributed to the difference in seed coat permeability properties as the seed coat is the main interface between the embryo and the external environment. Mayne et al. (1969) reported that hard and impermeable seed coat prevented moisture fluctuations in the embryonic axis and helps in prolonging the storability of seeds.

Physical structural changes in seed coat during aging

The seed coat in maize are called pericarp which are shown to have three layers i.e epidermis, hypodermis and the innermost layer containing remnants of thin-walled cells. The results of our study (Fig. 2) show that African Tall genotype with high EC had crushed innermost parenchymal layer while MAH 14-5 and local landrace that had low EC had intact innermost parenchymal layer. Thus the presence of an intact interior parenchymal layer might be the reason for impermeability and seed coat integrity in colored seed coat than in colorless seed coat of maize.

Also, studies have noticed the difference in the hypodermal cells. They are called hourglass cells, pillar cells, lagnoscelerids or osteoscelerids depending upon the cell wall thickness and shape (Souza & Marcos-Filho, 2001). However clear function, impermeability or seed coat strength imposed by different shapes of hypodermal cells are not studied. Our results (Fig. 2) show that in local landrace which had an intact and impermeable seed coat, the hypodermal cells were rectangular whereas in MAH 14-5 which showed intermediate permeability, the cells were elliptical while, in African Tall with a permeable seed coat had circular cells. This pattern of cell shapes shows that rectangular cells had less intracellular space while elliptical cells had comparatively more intracellular space and bit loosely packed while the circular cells had high intracellular space. This shows that cells in hypodermis had shapes which had less intracellular space has high impermeability however further studies need to be performed to establish a definitive relationship.

Besides this, Fig. 4 shows that African Tall seeds have a slightly curved rectangular pattern, MAH 14-5 had straight bordered rectangles while local landrace had a curved line pattern on the epidermis. Seed coat microsculpture is used as taxonomical diagnostic characters for distinguishing genera and species of heterogeneous families (Johnson, Huish & Porte, 2004). Thus the different microsculpture patterns observed between genotypes can be used in species differentiation. Also, the microsculpture pattern arises due to lignified thickening in the walls of the testa layer which shows the differential thickening and chemical composition between the three genotypes. SEM images also revealed that intact seed coat in fresh seeds of three genotypes had no damage however the damage in seed coat started to accumulate as the seeds aged (Fig. 5). However, the level of seed coat damage was least in aged seeds of local landrace followed by MAH 14-5 while African tall incurred increased damage. This shows that the colored seed coat (Local landrace and MAH 14-5) were comparatively strong and more resistant to aging-induced deterioration than colorless seed coat (African Tall) which might be due to the thickened seed coat as seen in microsculpture patterns. Figure 6 shows clearly that the pattern of damage is concerning the microsculpture pattern where the damage is along the microsculpture pattern. This clearly shows the influence of seed coat microsculpture patterns on seed coat damage among the genotypes.

Staining with Toluidine blue (Fig. 7B) revealed that phenols were present only in the epidermal and hypodermal layers of the seed coat while it was not present in the interior parenchyma layer. Figure 3 shows that initial damage due to aging starts from the interior parenchyma layer, thus the absence of phenols in this layer might be a possible reason for easier damage to this layer. Troszynska & Ciska (2002) have reported the antioxidant potential of phenols and their role in seed hardness. Thus, phenols present in the epidermis and hypodermis layer of maize seed coat might be imparting strength and hardness to seeds acting as a defense against seed coat damage.

Antioxidant enzyme activity in seed coat during aging

Higher catalase, superoxide dismutase, peroxidase and polyphenol oxidase activity were reported in the seed coat of colored genotypes (local landrace and MAH 14-5) that resisted aging and deterioration while, lower catalase, superoxide dismutase, peroxidase and polyphenol oxidase activity was reported in colorless African Tall seed coat that deteriorated during aging. Similar results of antioxidant activity in whole seeds were observed by Pawar et al. (2019).

When analysed with results of germination and vigour, colored genotypes deteriorated less as seeds aged compared to colorless genotypes. The linear regression analysis of catalase activity and germination revealed a strong positive relationship with R2 = 0.98, 0.94 and 0.72 in African Tall, MAH 14-5, Local landrace respectively (Fig. 9A). Thus, catalase is shown to reduce the free radical damage to seeds by scavenging activity thereby enhancing germination while, reduced catalase activity leads to seed deterioration. However, colored seed coat had comparatively higher catalase activity at any point of time during aging than colorless seed coat which might be a possible reason for the improved storability and vigour of colored seeds. Similar results of high catalase activity in fresh seeds and decreased activity in aged, deteriorated seeds in poor storer type and low catalase activity in fresh seeds and increased activity in aged seeds in good storer were observed by Hosamani et al. (2013), Dalurzo et al. (1997) and Bhattacharjee et al. (1998).

Figure 9 Simple linear regression analysis of germination percent and antioxidant enzyme activity in seed coat during different periods of ageing in maize genotypes.

(A) African Tall, (B) MAH 14-5, (C) local landrace.

The linear regression analysis of superoxide dismutase activity and germination revealed a strong positive relationship with R2 = 0.98, 0.97 and 0.77 in African Tall, MAH 14-5, Local landrace respectively (Fig. 9B). Thus, SOD is shown to reduce the free radical damage to seeds by scavenging activity as SOD is the first in the series of the antioxidant system that scavenges superoxide radicals produced during the electron transport process and thus reduced SOD activity leads to seed deterioration. Also, studies show that SOD plays a crucial role in the process of free radical scavenging and in seed aging (Hosamani et al., 2013).

The linear regression analysis of peroxidase activity and germination revealed a strong negative relationship with R2 = 0.99, 0.98 and 0.95 in African Tall, MAH 14-5, Local landrace respectively (Fig. 9C). The decreased peroxidase activity in seeds with aging reveals the antioxidant role of peroxidase which in turn might lead prevent seed deterioration.

Also, the polyphenol oxidase activity in the seed coat had a strong positive relationship with the concentration of phenol present in the seed coat with R2 = 0.95 between the three genotypes (Fig. 10). Earlier studies have reported that oxidized phenols impart impermeability to seed coats by a cross-linking of cell walls (Pourcel et al., 2005). Higher polyphenol oxidase activity in colored seed coats (local landrace and MAH 14-5) might be the reason for intact seed coats while African Tall was permeable and had damage to the seed coat. The relationship between phenol and permeability with phenol imposing impermeability is well established (Marbach & Mayer, 1974). This is further proved by TRANSPARENT TESTA 10 (tt10) mutants encoding a laccase-like enzyme involved in oxidative polymerization of phenols had less dormancy, and increased permeability compared to control (Debeaujon, Léon-Kloosterziel & Koornneef, 2000; Pourcel et al., 2005). The linear regression analysis results of the present study also reveal a strong negative relation between phenol concentration in seed coat and electrical conductivity among three genotypes with R2 = 0.96. This shows that phenol imparts strength to the seed coat and prevents membrane damage (Fig. 11).

Figure 10 Simple linear regression analysis of polyphenol oxidase activity and phenol concentration in seed coat of three maize genotypes.

Figure 11 Simple linear regression analysis of phenol concentration in seed coat and electrical conductivity of three maize genotypes.

Conclusion

The study demonstrates that there are differences in seed coat physical structure and antioxidant activity during accelerated aging among different genotypes. As the seeds mature, damage accumulates on the seed coat surface and the antioxidant activity decreases. These factors may contribute to the deterioration of seeds as they age due to the buildup of free radicals. Therefore, these findings clearly illustrate the impact of seed coat structure and antioxidant activity on seed quality during aging.

It is important to conduct similar studies in different genotypes under natural aging conditions to validate the results. This study has laid the groundwork for improving seed storability through selective breeding to enhance seed coat properties. Additionally, techniques such as seed coating, pelleting and priming can be used to strengthen the seed coat and enhance storability.

Supplemental Information

Supplemental Information 1 Antioxidant data

Supplemental Information 2 Figures 8–11 data

Supplemental Information 3 Germination and viability data

Supplemental Information 4 Vigour and moisture

Supplemental Information 5 TDH

The authors would like to acknowledge the Central Instrumentation Facility, Department of Crop Physiology, University of Agricultural Sciences, Bangalore for providing Scanning Electron Microscopy facility and Dr. Nataraja Karaba N, Principal Investigator for the ICAR-NAHEP-CAAST project on “Center for Next Generation Technologies for Adaptive Agriculture” (ICAR-CAAST-Project F.No./NAHEP/CAAST/2018-19), Dr. Shivsharan Patil and Hemanth, Department of Crop Physiology, University of Agricultural Sciences, Bangalore for their assistance and support in acquiring SEM images. We also thank Dr. MK Prasanna Kumar, Pathogenomics Lab, Department of Pathology, University of Agricultural Sciences, Bangalore for providing the Multispec facility and Prakash Diagnostic laboratory and Histotech services, Bangalore for providing the microtome facility.

Additional Information and Declarations

Competing Interests

Author Contributions

Data Availability

The authors declare there are no competing interests.

Vijayan Satya Srii conceived and designed the experiments, performed the experiments, analyzed the data, prepared figures and/or tables, authored or reviewed drafts of the article, and approved the final draft.

Nethra Nagarajappa conceived and designed the experiments, authored or reviewed drafts of the article, project supervision, funding acquisition, and approved the final draft.

The following information was supplied regarding data availability:

The raw data are available in the Supplemental Files.

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
