# Peer review of "Impact of accelerated aging on seed quality, seed coat physical structure and antioxidant enzyme activity of Maize (Zea mays L.)"

_PeerJ, doi:10.7717/peerj.17988_

## Round 0.1 · original submission · Major Revisions

Please address the concerns of all reviewers and amend the manuscript accordingly.

Reviewer 1 ·

Basic reporting

no comment

Experimental design

Methods described with sufficient detail & information to replicate

Validity of the findings

Impact and novelty not assessed. Meaningful replication encouraged where rationale & benefit to literature is clearly stated

Additional comments

Vijayan et al., …………..was conducted research on Impact of accelerated ageing on seed quality, seed coat physical structure and antioxidant enzyme activity of Maize (Zea mays L.). After review, I failed to find a novel presentation of the research conducted mainly in the form of manuscript documentation, figure clearness, and proper fit to establish the mechanistic of agricultural filed in strengthening the outcomes.

·

Basic reporting

Thank you for suggesting me to review the attached manuscript.
Through this article, authors tried to investigate the effect of accelerated ageing on the physical structure and activity of selective antioxidant enzymes of Maize seeds. However, after thoroughly reading the manuscript, I have to say this:
-Principally, I can’t understand why authors chose Maize! Is there a real issue considering seeds deterioration in maize? To my knowledge there is not. Alternatively, there are several crops knowing for their sensitivity to storage conditions especially, soybean and sunflower! The used references approved this, as most of it focused on soybean!
Abstract
Com. 1: Lines 9 and 43. Seed deterioration has many minor and major effects, but it is too early to propose such huge effect “genetic damage or changes”! We cannot assume that the noticed phenotypic alterations in a certain genotype is due to genetic changes that may resulted from seed deterioration!
Introduction
Com. 2: Some editing for English language is required throughout the manuscript due to too many mistakes. Too long statements have been used in several places in this manuscript, especially the discussion section that need to be summarized. Also, either US or UK English should be used, don’t mix!
Com. 3: This section must be rewritten and relevant up-to-date references should be included accordingly! Too long phrases are without citing other authors. It is not known who the authors write the preliminary assumptions for their research.
“The underlying rationale is that when seeds are subjected to adverse conditions (high temperature and RH), the same level of deterioration would occur in seeds as natural ageing but over a shorter period.” First of all, this assumption cannot be totally true, however references support these claims still needed. Lines: 35-40
Com. 4: These are long stories described without references? These should be completed with relevant up-to-date literatures. Lines 51-54, 58-64.
Com. 5: Introduction should be finished with scientific hypothesis well formulated, this manuscript has lack of this statement which is not acceptable.
Why those genotypes!! Is there any relationship between seed coat color and deterioration! This needs to be clarified. Line 79.

Experimental design

Materials & Methods
Com.6: Please, state in which season seed multiplication was occurred. Line 87.
Com. 7: Is the second reference relevant and significantly contributed to the development of the adopted approach? Line 92.
Com. 8: Authors have to follow the same approach in determining seed moisture, this will guarantee more accurate estimation! However, in Lines 134-136 authors stated that a moisture meter has been used to estimate seed moisture described as non-distractive method!! This is confusing!! Lines 93-102.
Com. 9: Authors have to clarify why they used only two aging durations of 96 and 120 hours? Additional aging levels will participate effectively in addressing this critical issue! Line 100.
Com. 10: Different number of replicates have been used, why? Few replicates will resulted in bias evaluation! Lines 106-112.
Com. 11: Spectrophotometer manufacturer should be stated. Line 142.
Com. 12: Three or two genotypes were included! However, results and discussion refers to three genotypes in TDH test! Line 144.
Com. 13: More details about sample preparation for SEM examination should be mentioned, including the manufacturer and model of SEM device. Line 157.

Validity of the findings

Results and discussion
Com. 14: The used equation for assaying Polyphenol oxidase activity needs to be corrected. Line 215
Com. 15: The following statement is part of Materials and Methods “ANOVA with two 274 factors was used to test the significance of the data and there was a significant difference 275 (P<0.01) in seed quality parameters between genotypes and between different ageing treatments”. Lines 273-275
Com. 16: I have doubts that the detected difference in EC values between the three used genotypes was significant. Please, make sure the following statement is true “but there was a difference in EC values between genotypes”! Line 270.
Com. 17: This statement should be moved to the discussion section “Previous studies show that TDH content decreases in aged and deteriorated seeds (Verma et al., 2003)”. Lines 294-295.
Com. 18: This paragraph is a basic knowledge, no need for such general information in results section. Lines 300-307.
Com. 19: Don’t go beyond your findings, there is no need for such long general statement. Lines 415-423.
Com. 20: I don’t think correlation will support the already explained findings. Lines 415-423.
Com. 21: In general, most of the used images are not in a good quality, several concerns need to addressed. For example, different resolution can be detected in a single figure, different pattern of seed coat in each genotype can be easily noticed, some images captions are not informative and need further clarification (e.g. light microscope or SEM). Also, the detected cracks looks normal and common in crops seeds, and could be occurred naturally or during harvesting process and/or sample preparation. A clear variation in seed coat microsculpture pattern even in a single genotypes can be observed!

Additional comments

I wish authors will able to use these comments to improve their article before re-submission.
Best regards

·

Basic reporting

Needs language editing

Experimental design

Appropriately designed

Validity of the findings

No comment

Additional comments

Landrace name may be mentioned.
The reason for colouring in Landrace may be provided.
Lines 13-16 shall be revised.
Lines 31-33, references needs to be placed chronologically.
Lines 54-55, the statement is not clear.
Lines 79-80, may be removed since it has been given again in Lines 83-84 of Materials and Methods section.
Lines 110-114, protocol already available in ISTA guidelines, thus provide ISTA reference. Remove another reference.
Lines 122-125, procedure for obtaining seedling dry weight needs to be given.
Lines 134-136, hot air oven method gives accurate result than moisture meter (make and model not provided).
Lines 144-145, may be removed and provided in statistical analysis part.
Lines 161-169, language editing is needed.
Line 171, may be removed.
Line 215, check the formula.
Lines 236-242, needs revision, language editing.
Lines 243-244, needs revision, language editing.
Lines 261-262, shall be removed and placed under the germination test result.
Lines 365-366, statement says that the polyphenol oxidase activity showed no significant changes with ageing times in all genotypes but graph shows slight change in polyphenol activity. It needs verification, whether the data were statistically analysed and got Non-significant values. If it is so, it has no role in protecting the seed from deterioration. It needs clear justification along with the statements provided in Lines 494-507.

Reviewer 4 ·

Basic reporting

1. The English language and overall grammar of the manuscript need improvement for clarity and readability.
2. Figures 9-11 do not show correlation. The graphs indicate the regression equations, and R² represents the coefficient of determination. Please correct these graphs and the corresponding text in the manuscript.

Experimental design

1. The introduction provides a good overview of the deteriorative changes in seeds due to ageing. However, it would benefit from a clearer statement of the research objectives and hypotheses. Consider explicitly stating the main aims of the study at the end of the introduction.
2. Line 84: Were all three genotypes fodder maize? If so, was there a specific reason for choosing fodder maize genotypes for the study instead of grain maize genotypes?
3. Ensure consistent use of abbreviations throughout the manuscript.
Authors should format all equations correctly using an appropriate equation editor; they are not in the correct format currently.
4. Line 243: Was the initial germination percentage 100% for all the genotypes? This seems unrealistic.

Validity of the findings

Authors should structure the discussion section by assigning separate sections to each objective as described in the introduction. Additionally, the potential practical implications of the findings for seed storage and preservation should be elaborated upon.
Authors should provide a separate section for the conclusion. Consider adding a brief statement on future research directions or practical applications of this work in the conclusion.

---

## Round 0.2 · Minor Revisions

Please address the concern of the reviewer related to the statement that the scale bar of 20 μm was used for all images, as it seems that this statement is not applicable to the Local Red Landrace-Control image.

·

Basic reporting

Thank you again for suggesting me to review the attached manuscript.
However, after thoroughly reading the revised manuscript, I'm convinced that the authors has made most of the required amendments, and that the manuscript in its current state is suitable for publication.
- English language was improved thoroughly.
- Up-to-date and relevant literature references were provided.
- Figure 5: It was stated that the used Scale bar is 20 μm for all images! However, this is Not Applicable for Local Red Landrace-Control, it is easy to detect low resolution of this image compared to the other images!
Best regards

Experimental design

This section was improved by the authors according to the previous reviewing report.

Validity of the findings

Authors considered the previous reviewing report and improved this section accordingly.

Additional comments

No additional comments

---

## Round 0.3 · accepted · Accept

Remaining issues indicated by the reviewer were adequately addressed and amended manuscript is acceptable now.